# Enhancer regions show high histone H3.3 turnover that changes during differentiation

Aimee M Deaton[1,2†], Mariluz Gómez-Rodríguez[3], Jakub Mieczkowski[1,2], Michael Y Tolstorukov[1,4], Sharmistha Kundu[1,2], Ruslan I Sadreyev[1,5], Lars ET Jansen[3*], Robert E Kingston[1,2*]

[1]Department of Molecular Biology, Massachusetts General Hospital, Boston, United States; [2]Department of Genetics, Harvard Medical School, Boston, United States; [3]Laboratory for Epigenetic Mechanisms, Instituto Gulbenkian de Ciencia, Oeiras, Portugal; [4]Department of Medicine, Harvard Medical School, Boston, United States; [5]Department of Pathology, Massachusetts General Hospital and Harvard Medical School, Boston, United States

**Abstract** The organization of DNA into chromatin is dynamic; nucleosomes are frequently displaced to facilitate the ability of regulatory proteins to access specific DNA elements. To gain insight into nucleosome dynamics, and to follow how dynamics change during differentiation, we used a technique called time-ChIP to quantitatively assess histone H3.3 turnover genome-wide during differentiation of mouse ESCs. We found that, without prior assumptions, high turnover could be used to identify regions involved in gene regulation. High turnover was seen at enhancers, as observed previously, with particularly high turnover at super-enhancers. In contrast, regions associated with the repressive Polycomb-Group showed low turnover in ESCs. Turnover correlated with DNA accessibility. Upon differentiation, numerous changes in H3.3 turnover rates were observed, the majority of which occurred at enhancers. Thus, time-ChIP measurement of histone turnover shows that active enhancers are unusually dynamic in ESCs and changes in highly dynamic nucleosomes predominate at enhancers during differentiation.

*For correspondence: ljansen@ igc.gulbenkian.pt (LETJ); kingston@molbio.mgh.harvard. edu (REK)

Present address: †Amgen Inc, Cambridge, United States

Competing interests: The authors declare that no competing interests exist.

## Introduction

The organization of the genome into chromatin acts as a mechanism for regulating access to the information encoded in DNA. The simplest repeating unit of chromatin is the nucleosome consisting of a histone octamer, made up of two copies each of H2A, H2B, H3 and H4, around which approximately 147 bp of DNA is wrapped. Variants of the histone proteins exist that have distinct incorporation dynamics and functions. These include a number of histone H2A variants such as H2A.Z, H2A. X and macroH2A and histone H3 variants such as H3.3 and CENP-A (*Maze et al., 2014*; *Skene and Henikoff, 2013*).

As nucleosomes can act as barriers to reading the DNA sequence, they must be disrupted in order for many processes requiring DNA access to occur. Transcription and the binding of proteins to regulatory regions are known to be dependent upon increased accessibility to nucleosomal DNA. Therefore, assessment of nucleosome turnover can provide important information concerning which genomic regions are involved in regulation and the extent to which nucleosome dynamics contribute to their function.

Nucleosome dynamics is a generally underexplored property of chromatin. Previous studies examining nucleosome turnover genome-wide have made use of two main strategies – induction of

**eLife digest** In animal, plant and other eukaryotic cells, DNA wraps around histone proteins to form structures called nucleosomes. This compacts long strands of DNA to fit them inside a cell. However, nucleosomes also act as barriers that can prevent access to the DNA. This affects the activity, or "expression", of genes because gene expression requires proteins called transcription factors to bind to specific DNA regions. Therefore, nucleosomes must be disrupted or removed in order to access their DNA and allow their genes to be expressed.

Transcription factors can bind to DNA sequences called enhancers to activate nearby genes. Groups of enhancers, called super-enhancers, also exist to further bolster the activity of certain genes, particularly those involved in determining cell identity. Recent work has shown that nucleosomes are frequently lost and then replaced by new ones (in a process referred to as turnover) in DNA regions that include enhancers. Measuring the rate of turnover of nucleosomes can thus provide information about which DNA regions regulate gene expression.

Embryonic stem cells can transform or "differentiate" into any type of cell in the body. During this transformation process, different genes are switched on or off in the cell in order to give it a new identity. It is not known how nucleosome turnover changes when this happens.

Deaton et al. have now developed a new method called time-ChIP that can measure the rate of nucleosome turnover across the entire DNA of a cell. Using this technique to analyze mouse embryonic stem cells revealed that nucleosome turnover occurs rapidly at enhancers. Furthermore, nucleosomes at super-enhancers are particularly dynamic and turn over more quickly than in any other DNA region.

Deaton et al. next analyzed how turnover changes after the mouse embryonic stem cells have developed into neural stem cells. This revealed that the regions of DNA where high turnover occurs change as the cells differentiate, in part because this transformation activates a different set of enhancers. However, the most rapid turnover still takes place at enhancers.

Overall, these observations suggest that the high rate of nucleosome turnover at enhancers makes DNA accessible to transcription factors. The next step is to use the new time-ChIP method to study how nucleosome turnover changes during the processes that pattern gene expression as an animal develops from an embryo.

epitope-tagged histone transgenes (*Kraushaar et al., 2013*; *Yildirim et al., 2014*) and metabolic labeling of histone proteins (*Deal et al., 2010*). These have yielded a number of insights into the genome-wide pattern of nucleosome turnover in Drosophila and mammalian systems. Such insights include high nucleosome turnover at gene bodies, regulatory elements and replication origins in Drosophila cells (*Deal et al., 2010*) and, in mouse cells, rapid deposition of H3.3-containing nucleosomes at enhancers and promoters associated with active histone modification marks and the histone variant H2A.Z (*Kraushaar et al., 2013*). However, each set of techniques has limitations. For example, systems based on inducible histones have a lag time after induction before histones are synthesized and incorporated into chromatin. In addition, they mainly look at where newly synthesized histones are deposited rather than disruption of existing histones. A TET-OFF system examining dissociation of tagged histones has recently been described although a significant proportion of tagged histone is observed up to 12 hr after addition of doxycycline, limiting the sensitivity of this technique (*Ha et al., 2014*). Metabolic labeling methods such as 'CATCH-IT', on the other hand, are not specific for particular histone variants and have limited resolution at the level of individual genes. CATCH-IT analyzes incorporation rates rather than existing histones and cannot be employed for long chase times.

To circumvent these limitations, we have employed a newly developed method called time-ChIP (Gómez-Rodríguez et al., submitted) to assess histone turnover genome-wide during differentiation of mouse embryonic stem cells (ESCs) to neural stem cells (NSCs). Time-ChIP is based on SNAP technology (*Bodor et al., 2012*; *Keppler et al., 2003*), where histones are tagged with the suicide enzyme SNAP. Pulse labeling with a SNAP-tag specific biotin moiety leads to covalent biotinylation of the SNAP-tagged histone pool. Histone dynamics can then be determined by following the loss

of biotin-labeled histones over time as nucleosomes are disrupted and replaced with nucleosomes comprised of newly synthesized unlabeled histones. Sequencing of the DNA associated with these biotinylated histones allowed us to quantify relative histone turnover genome-wide. Previous studies using SNAP-tagged histones suggest that they are recognized by the correct chaperone proteins and incorporated into chromatin at the appropriate cell cycle stage (*Bodor et al., 2013*; *Dunleavy et al., 2009*; *Foltz et al., 2009*; *Jansen et al., 2007*; *Ray-Gallet et al., 2011*).

Time-ChIP allows labeling of existing, incorporated histones, is specific for particular histone variants and allows us to study histone dynamics in vivo. We focused on measuring turnover of the histone variant H3.3. In contrast to canonical H3, histone H3.3 is deposited independently of replication at places where nucleosomes are disrupted such as promoters, the bodies of active genes, and regulatory elements (*Goldberg et al., 2010*; *Schwartz and Ahmad, 2005*). Therefore examining this variant is likely to yield insights into the dynamics at regions important for gene regulation.

To examine H3.3 turnover at developmental loci and regulatory elements, and understand how turnover changes during cell specification, we applied time-ChIP to neural differentiation of mouse ESCs. We found that time-ChIP could be used a priori to identify regions important for gene regulation; specifically, enhancer regions stood out for their rapid turnover. In ESCs, consistent with previous work, we found high H3.3 turnover at active enhancers (*Ha et al., 2014*; *Kraushaar et al., 2013*) with even higher rates of turnover observed at super-enhancers. We also observed high turnover of histone H3.1 at these regions. Regions bound by Polycomb-group proteins, which are involved in developmental gene repression, showed H3.3 enrichment in ESCs but low turnover compared to active regions. Upon differentiation to NSCs, many changes in H3.3 turnover were observed, with ESC and NSC enhancers overrepresented in the set of regions showing changes. We corroborated our time-ChIP findings by correlating H3.3 turnover with DNA accessibility measured by micrococcal nuclease (MNase) titration (*Mieczkowski et al., 2016*) and found good correlation at enhancers and other active regions. Our findings demonstrate the utility of time-ChIP for profiling H3.3 turnover during differentiation, for determining relative rates of turnover at regulatory elements, and for characterizing known and potentially novel regulatory regions.

## Results

### Time-ChIP reports histone H3.3 turnover genome-wide

We used time-ChIP to assess histone H3.3 turnover genome-wide in ESCs in an unbiased manner. To do this, we used a SNAP-tagged histone H3.3 which can be labeled with fluorescent or biotin-containing substrates in cell culture (*Bodor et al., 2013*; *Ray-Gallet et al., 2011*). We generated ESC lines expressing SNAP-tagged H3.3 in single copy from a defined locus. To achieve this we used recombination-mediated cassette exchange to insert H3.3 C-terminally tagged with SNAP and HA-tags into A2lox-Cre ESCs which allow for dox inducible transgene expression from the *hprt* locus (*Iacovino et al., 2011*) (*Figure 1—figure supplement 1*). Western blotting showed that H3.3_SNAP was expressed at low levels relative to total endogenous H3 (*Figure 1A*) and undetectable when blotting was performed using an H3.3-specific antibody (*Figure 1—figure supplement 1*). This was desirable, as we wanted H3.3_SNAP to act as a tracer to report the turnover of all H3.3 rather than overwhelm the cells with high levels of transgenic histone. We also generated a line expressing low levels of SNAP-tagged histone H3.1 (*Figure 1—figure supplement 2*). Post-translational modifications were detected on the SNAP-tagged histones (*Figure 1—figure supplement 2*).

ESCs expressing SNAP-tagged H3.3 were used to perform time-ChIP and assess genome-wide H3.3 turnover at high resolution. After inducing tagged H3.3 for 48 hr, H3.3_SNAP was biotin labeled in intact cells by adding the cell permeable molecule chloropyrimidine biotin (CP-biotin) (*Correa et al., 2013*) to the media for 40 min followed by a wash step to remove unbound substrate. A portion of the labeled cells was harvested immediately representing the 0 hr time point and the remaining cells were re-plated for harvest at 3 hr, 6 hr and 12 hr post-labeling. Nuclei were isolated from these cells, chromatin liberated by MNase digestion and biotin labeled H3.3 recovered by streptavidin pull-down. DNA was purified from the recovered nucleosomes, quantified and prepared for sequencing (*Figure 1B*). As expected, decreasing amounts of DNA were recovered over time as biotin labeled H3.3_SNAP was replaced by newly synthesized H3 (*Figure 1C*). Approximately the

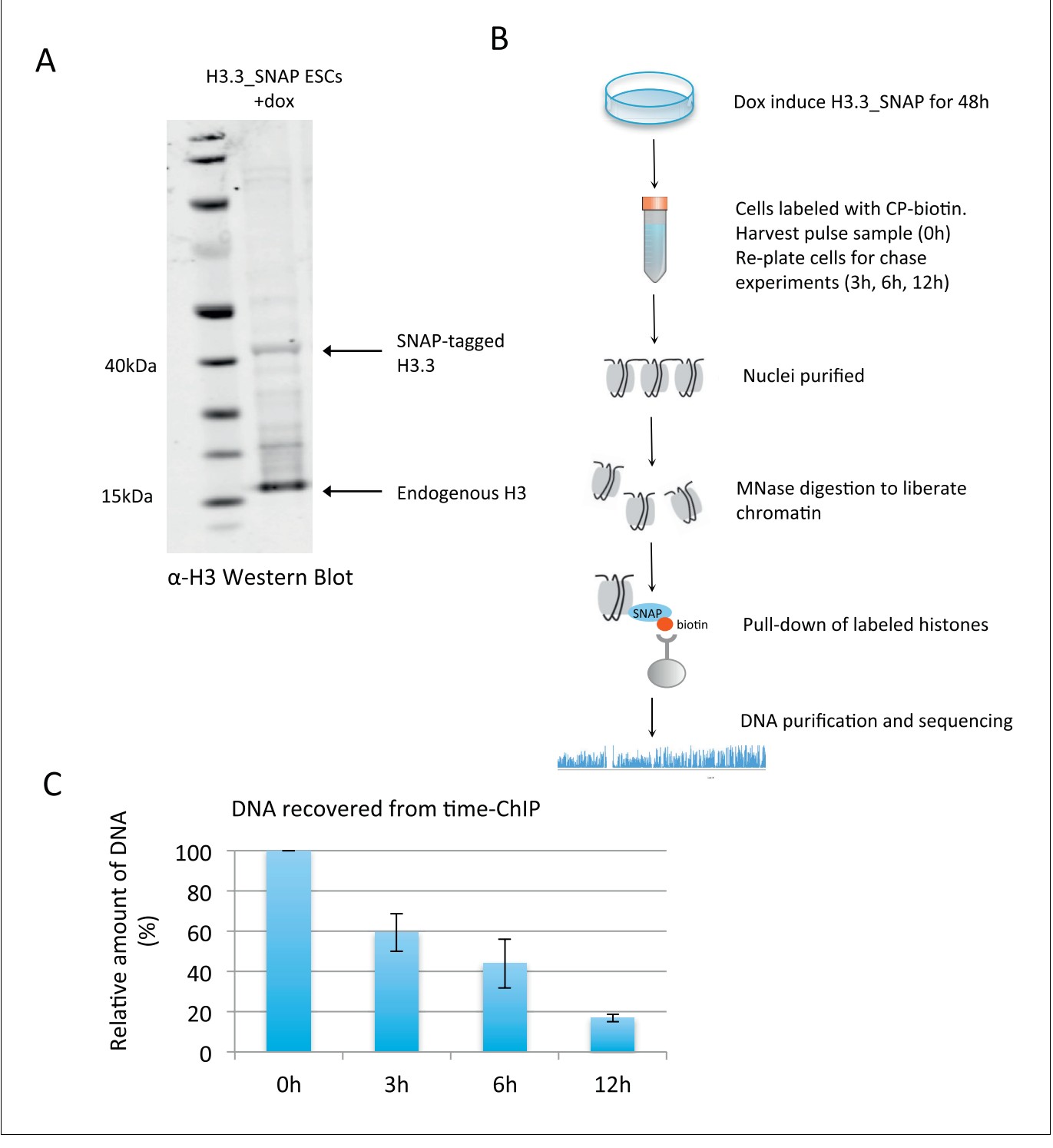

**Figure 1.** Time-ChIP assay in H3.3_SNAP ESCs. (**A**) Western blot for total H3 in H3.3_SNAP ESCs after 48 hr dox induction of transgene. (**B**) Outline of time-ChIP protocol. (**C**) Quantification of DNA recovered from time-ChIP experiments. The amount of DNA relative to the 0 hr sample (expressed as %) was averaged over two H3.3 time-ChIP replicates. Error bars are ± standard deviation.

The following figure supplements are available for figure 1:

**Figure supplement 1.** Time-ChIP assay in H3.3_SNAP ESCs.

**Figure supplement 2.** Post-translational modification of SNAP-tagged histones.

*Figure 1 continued on next page*

*Figure 1 continued*

**Figure supplement 3.** Spike-in control.

same number of reads was sequenced from each time point. A spike-in control was added when preparing the sequencing libraries to verify that the proportion of reads resulting from pull-down of SNAP-tagged histones was consistent with expectation for each time point. Later time points in each experiment had fewer fragments associated with tagged histones compared to spike-in fragments and thus were consistent with the expected overall H3.3 turnover rate in the genome (*Figure 1—figure supplement 3* and Materials and methods).

To validate that the SNAP tagged histone showed appropriate localization, we analyzed the 0 hr sample. As expected, H3.3 was enriched at enhancers and the promoters and bodies of active genes. H3.3 also showed high enrichment at transcription termination sites (TTSs) (*Figure 2—figure supplement 1*). To ensure that the biotin labeling of H3.3 was representative of all tagged H3.3, we compared the enrichment profiles for the H3.3 0 hr sample to those generated by performing HA ChIP for transgenic H3.3_SNAP_HA and the two correlated well (*Figure 2—figure supplement 1*).

To further validate our approach we examined the transcription start sites (TSSs) of active genes as these are elements where one would expect high H3.3 turnover. We plotted H3.3 time-ChIP data for the 0 hr, 3 hr, 6 hr and 12 hr time points over the transcription start sites (TSSs) of active genes (*Figure 2A*). As MNase was used to digest the chromatin, a characteristic MNase digestion pattern was observed consisting of a nucleosome 'depleted' region at the TSS and a well-positioned +1 nucleosome. As time progressed, labeled H3.3 was lost from these TSSs at an appreciable rate, particularly upstream of the TSS, as indicated by the fact that the profiles for each time point were well spaced apart. In contrast, at the TSSs of silent genes, H3.3 was less enriched and the profiles for each time point were closer together indicating slower H3.3 turnover compared to active promoters (*Figure 2A*). Fast H3.3 turnover at the TSSs of active genes was evident at the level of individual loci as well as when average behavior was observed (*Figure 2B*). H3.3 enrichment and rapid turnover was also observed at the 3' ends of expressed genes (*Figure 2B and C*). These observations are consistent with the displacement of nucleosomes by the transcription machinery and the enrichment of the replacement H3 variant, H3.3, at sites of active transcription as reported previously (*Goldberg et al., 2010*). The timeframe for turnover of H3.3_SNAP is concordant with previous estimates based on incorporation of H3.3 (*Kraushaar et al., 2013*).

Our data allows assessment of the relative H3.3 turnover between different genomic sites. Regions of high turnover will have fewer reads in later time points than regions with slower turnover when compared to the number of reads seen for these regions at 0 hr. To examine how turnover compared between different regions, it was necessary to develop a method to quantify turnover rate. To do this we used data from the 0 hr, 3 hr, 6 hr and 12 hr time points to calculate a 'turnover index'. The genome was divided into 1 kb bins and, for each bin, signal from each time point was fitted to a simple linear regression model. The slope of the linear regression was weighted to account for significance of the data fit by the linear model and multiplied by minus 1 (*Figure 2D* and Materials and methods). The resulting value was taken as the turnover index (TI) such that regions with fast H3.3 turnover had higher positive TI values. Indeed, at the TSSs of active genes, TI was highest upstream of the TSS reflecting what we observed when individual time points were examined (*Figure 2E*). We calculated TI for three individual biological replicates and these samples correlated well with each other having R values of 0.67–0.82 (*Figure 2—figure supplement 1*). Therefore, for subsequent analyses, we used the mean TI of the three replicates as the TI measurement. This TI reflects relative H3.3 turnover between different genomic sites and, by its nature, is internally comparable across the genome.

Having established time-ChIP as a method for assessing H3.3 turnover and shown that it offers reproducible measurements genome-wide we then went on to examine H3.3 turnover in ESCs in more depth.

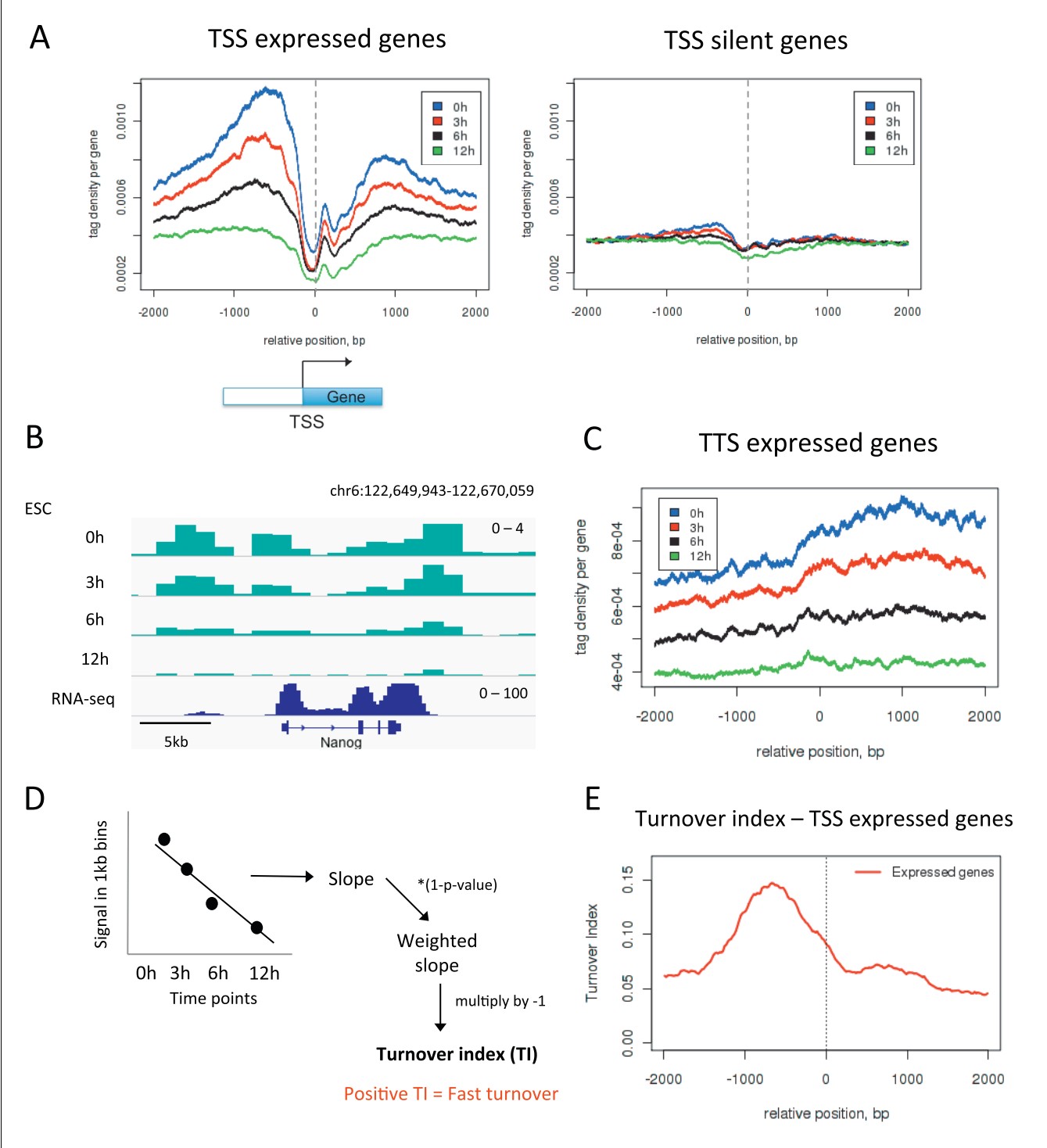

**Figure 2.** Measuring H3.3 turnover in ESCs. (**A**) Average H3.3 profiles for 0 hr, 3 hr, 6 hr and 12 hr time points over the transcription start sites (TSSs) of expressed and silent genes in ESCs. (**B**) H3.3 in 1kb bins profiles for 0 hr, 3 hr, 6 hr and 12 hr time points at the actively transcribed *Nanog* locus. (**C**) Average H3.3 profiles over the transcription termination sites (TTSs) of genes expressed in ESCs. (**D**) Calculating turnover index (TI) in 1 kb bins across the genome. H3.3 signal for each bin was plotted for every time point, these were fitted to a linear regression model and the slope of this line was weighted and multiplied by minus 1 to get TI. (**E**) Average TI over the TSSs of expressed genes. For (**A–C**) results from one experiment representative of three biological replicates is shown.

The following figure supplement is available for figure 2:

*Figure 2 continued*

**Figure supplement 1.** H3.3 enrichment in ESCs and TI correlation between replicates.

## Time-ChIP identifies regulatory regions in ESCs

We were interested in how turnover relates to regulation, so, without prior assumptions, we used TI to pick out the 1000 1 kb bins with the highest H3.3 turnover in ESCs (False Discovery Rate (FDR) = 0.003) (*Supplementary file 1*). The majority of these (approximately 77%) were located in at least one annotated regulatory region such as a promoter, gene body, TTS or enhancer. High TI regions were overrepresented, as compared to the occurrence of the features in the genome, at promoters, the 5' end of genes, TTSs, and enhancers (*Figure 3A*). In particular, high H3.3 enrichment and TI occurred at TTSs, often at those close to other transcription units, (*Figure 3B*) and at enhancers and super-enhancers (*Figure 3C*). We examined the high TI regions that did not correspond to annotated genomic features in more detail. Some of these overlapped with a larger UCSC gene set, leaving 146 high TI regions in truly unannotated locations. It is likely that a portion of these represents unidentified regulatory regions or regions with regulatory functions in differentiated cells (see example in *Figure 3D*). Indeed, 95 of these 146 unannotated regions were located within 50 kb of genes robustly expressed (FPKM > 9.5) in ESCs (p-value = 0.001) suggesting potential regulatory roles for them in ESCs. Thus, without prior assumptions, time-ChIP identifies known regulatory, and also potential novel regulatory regions. We conclude that time-ChIP can be used to identify regions important for gene regulation.

If TI changes with the nature of a regulatory region, we anticipated that TI would be high in regions of high expression when these were examined over the entire genome. TI in fact did correlate with gene activity and was higher over the promoters, bodies and TTSs of highly expressed genes compared to genes with little or no expression (*Figure 3E*). ESC enhancers showed particularly high TI values. This was consistent with their overrepresentation in the set of regions with the highest TI in ESCs, and further indicated that enhancers are especially dynamic.

We examined enhancers in more detail, and found that TI distinguishes isolated enhancers from the grouped enhancers that modulate expression of genes involved in cell identity; super-enhancers or locus control regions (*Magram et al., 1985*; *Whyte et al., 2013*). Plotting average TI profiles over conventional ESC enhancers and ESC super-enhancer constituents revealed higher H3.3 TI over the constituent enhancers that make-up super-enhancers domains compared to conventional enhancers despite the fact that these regions are similar in size (*Figure 3F*). When TI was calculated using a smaller 300 bp bin size, exceptionally high H3.3 turnover was still evident at super-enhancers (*Figure 3—figure supplement 1*). We also examined turnover of histone H3.1 at ESC enhancers and super-enhancers by performing H3.1 time-ChIP in the same manner as was done for H3.3. Although the depth of sequencing required to interrogate H3.1 limits the resolution of H3.1 time-ChIP, we clearly saw elevated TI over enhancers compared to randomized regions in two seperate biological replicates (*Figure 3—figure supplement 2*). In addition, super-enhancer constituents showed higher H3.1 turnover than conventional enhancers, consistent with the results for H3.3. The high H3.1 and H3.3 TI observed at super-enhancers suggests that nucleosomes are more frequently disrupted at these regions regardless of which H3 variant they contain. We conclude that TI offers a metric for identifying enhancers and super-enhancers, and that super-enhancers have an unusually dynamic nucleosome population.

## PcG-repressed loci show H3.3 enrichment but reduced TI in ESCs

Having observed increased TI associated with activation, we were interested in examining regions associated with repression. Heterochromatin, the most extensively repressed segment of the genome, does not have H3.3 (*Figure 4—figure supplement 1*), indicating a lack of turnover and causing us to be unable to measure H3.3 turnover at these regions. However Polycomb-Group (PcG) regulated genes are bound by H3.3 in ESCs (*Figure 4—figure supplement 1*), although H3.3 levels at these genes were lower than at enhancers and CpG islands (*Figure 4—figure supplement 1*). PcG proteins mediate repression, at least in part, through chromatin compaction (*Simon and*

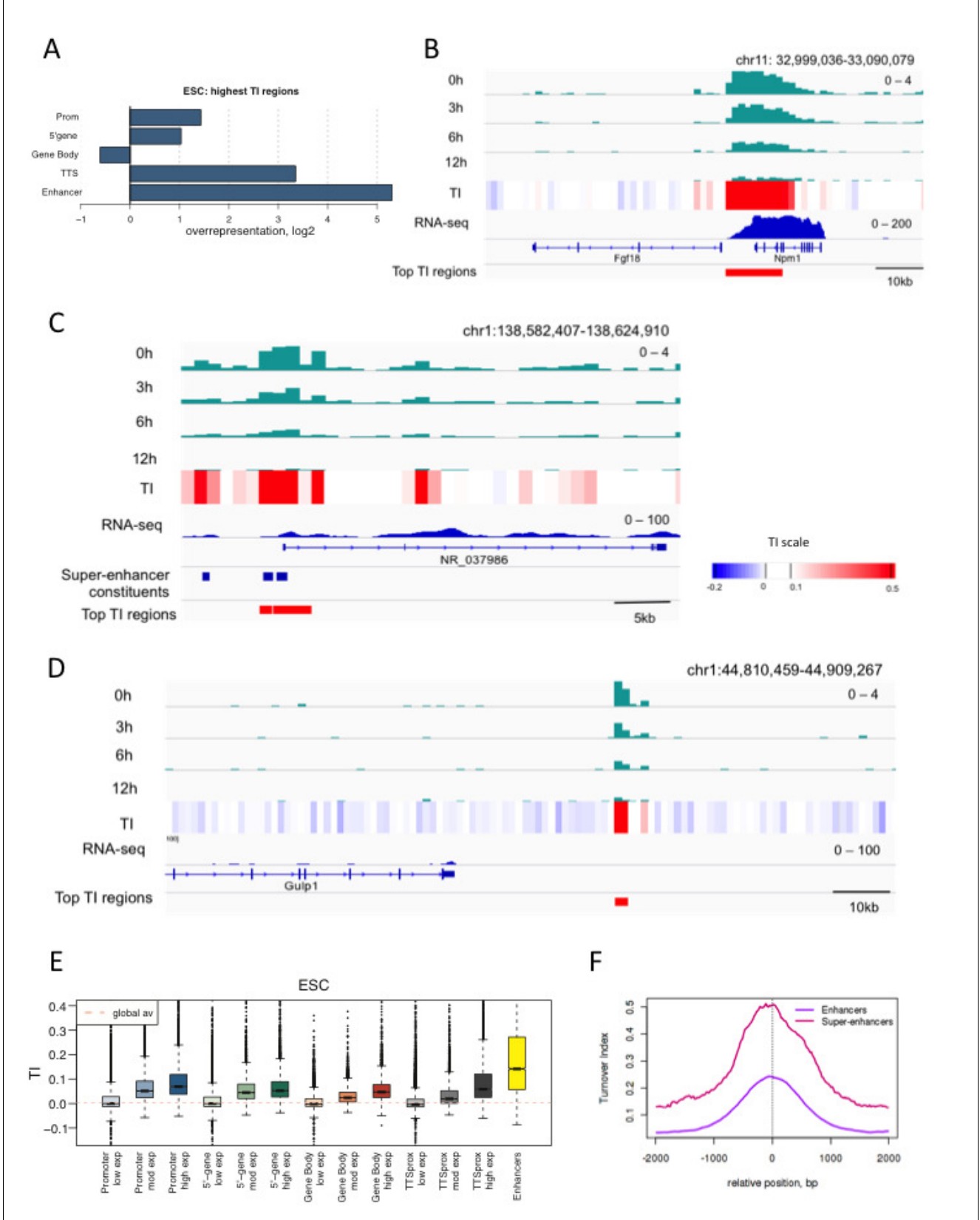

**Figure 3.** Regions with high TI in ESCs have regulatory functions. (**A**) Representation of annotated genomic features in the 1000 highest TI regions in ESCs relative to their representation in the genome as a whole. Overrepresentation was calculated by taking the log2 ratio of the proportion of the
*Figure 3 continued on next page*

*Figure 3 continued*

high TI regions overlapping a particular feature divided by the proportion of whole genome overlapping the same feature. (**B**) The region 5' of the *Fgf18* gene and 3' of the neighboring *Npm1* gene has high TI in ESCs. Signal for each time point is shown for one replicate (top 4 tracks – green) while TI averaged for 3 replicates is shown (heatmap). RNA-seq in ESCs is shown in blue while the highest TI regions are marked with red bars. (**C**) Super-enhancers in the *NR_037986* locus are amongst the most highly turned over regions in ESCs. Super-enhancer constituents are as defined by (***Whyte et al., 2013***) and shown as blue bars. (**D**) Unannotated regions are also present in the set of highest TI regions. The red bar marks a high TI region downstream of the *Gulp1* gene on chr1. (**E**) ESC TI distribution at annotated genomic elements – promoters, 5' end of genes, gene bodies, TTSs and enhancers. Genes are grouped by ESC expression level and ESC enhancers are from ***Whyte et al. (2013)***. (**F**) Average TI over conventional ESC enhancers (purple) and super-enhancer constituent enhancers (pink) as described in ***Whyte et al. (2013)***.

The following figure supplements are available for figure 3:

**Figure supplement 1.** H3.3 turnover at enhancers and super-enhancers calculated using a 300 bp bin size.

**Figure supplement 2.** H3.1 turnover at ESC enhancers and super-enhancers.

*Kingston, 2013*). One outstanding question is whether this compaction could lead to stabilization of nucleosomes and reduced nucleosome turnover at PcG-bound genes.

Measurement of TI at PcG-bound regions revealed slower turnover of H3.3 compared to all CpG islands, the majority of which are associated with active chromatin (***Figure 4A***) while PcG-bound TSSs had a TI intermediate between the TSSs of active and silent genes (***Figure 4B***). The classical PcG-target, the *Hoxb* locus shows low levels of H3.3 throughout the region with average TI values of 0.03 (***Figure 4C***), compared to high levels of H3.3 enrichment and an average TI of 0.14 in the active *Hnrnpa3* locus (***Figure 4D***). This shows that, in ESCs, PcG-bound regions have decreased TI compared to active regions.

Overall, these data indicate that histone turnover can relate to known regulatory features. Regions involved in activation have high TI, those involved in repression have low TI.

## High H3.3 turnover at enhancers and active genes after neural differentiation

ESCs have been proposed to have a more plastic chromatin structure compared with more differentiated cells (***Bernstein et al., 2006***; ***Meshorer et al., 2006***). Therefore it was possible that the high turnover we had observed at enhancers and super-enhancers was due to ESC-specific plasticity. We therefore determined whether turnover remained high at regulatory regions and enhancers following cellular differentiation.

We generated neural stem cells (NSCs) expressing H3.3_SNAP by injecting H3.3_SNAP ESCs into blastocysts, harvesting E13.5 embryos and deriving NSCs from embryonic brain. NSC lines were screened to identify transgenic lines (***Figure 5—figure supplement 1***) and these lines were drug-selected to obtain pure populations of cells carrying the H3.3_SNAP transgene. H3.3_SNAP NSCs expressed neuronal markers (***Figure 5—figure supplement 1***) and, upon dox induction, expressed H3.3_SNAP at similar levels to the ESC line (***Figure 5—figure supplement 2***).

To characterize turnover in neural stem cells, time-ChIP was performed with cells harvested at 0 hr, 3 hr, 6 hr and 12 hr post-labeling. As expected, genes that were activated during neural differentiation and lost PcG binding showed a dramatic increase in H3.3 enrichment and TI (***Figure 5A and B***). Regions still bound by PcG in NSCs were no longer enriched in H3.3 as they were in ESCs (***Figure 5C and D***). This suggests nucleosome stabilization at these regions upon differentiation. ESC-specific enhancers showed reduced H3.3 and TI in NSCs compared to ESCs but still showed high H3.3 turnover compared to the rest of the genome (***Figure 5E*** and ***Figure 5—figure supplement 2***).

Similar to what we observed in ESCs, the promoters, gene bodies and TTSs of highly expressed genes show high TI in NSCs as did putative NSC enhancers (identified using H3K27ac ChIP-seq data from embryonic brain). Average TI values were lower for putative NSC enhancers than those observed for active enhancers in ESCs (***Figure 5F***). This might reflect inclusion of enhancers expressed in embryonic brain that are not expressed in NSCs. When the 1000 regions showing the highest TI in NSCs (FDR = 0.002) were examined (***Supplementary file 2***), TTSs and enhancers were

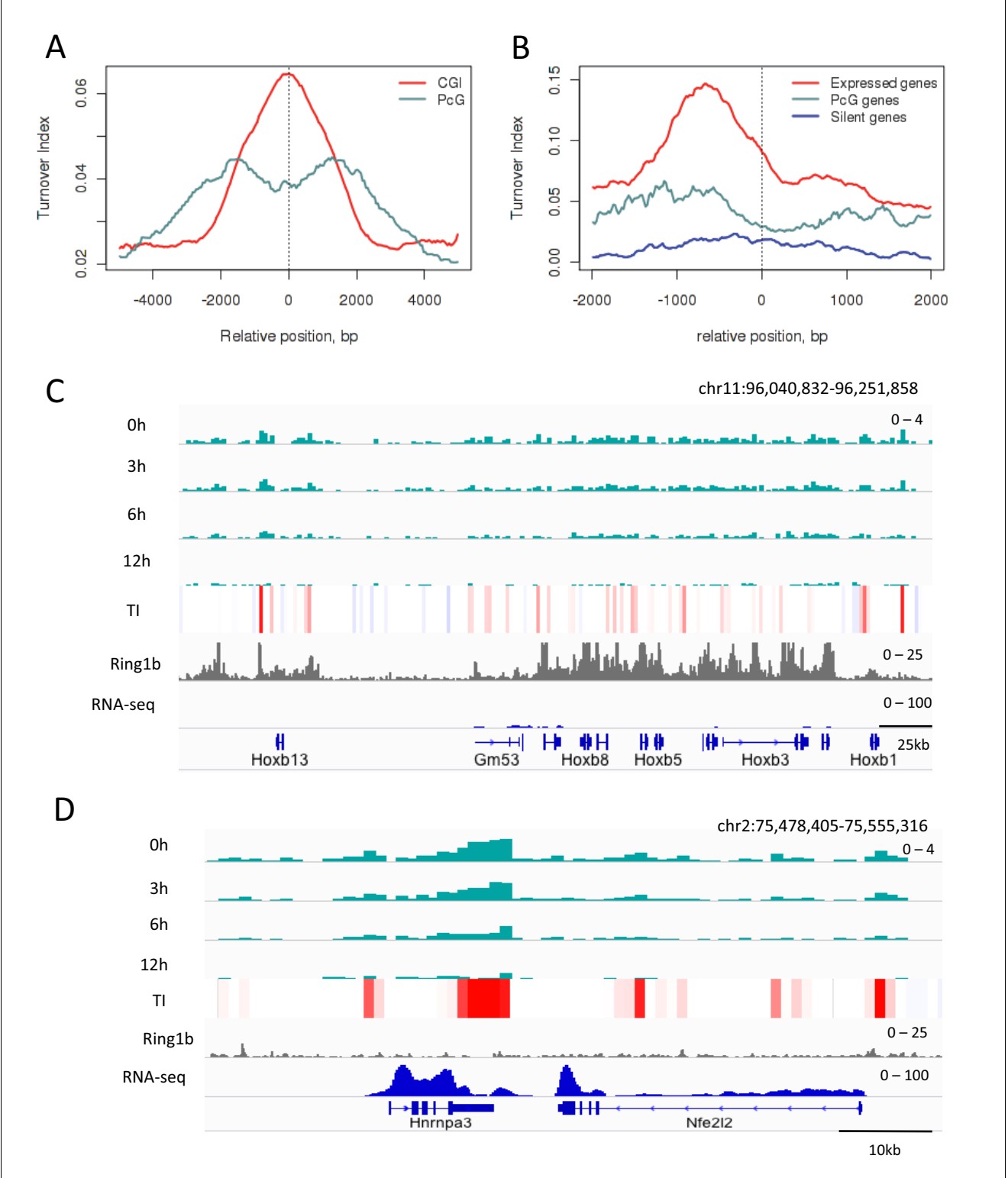

**Figure 4.** H3.3 turnover at PcG targets in ESCs. (A) TI averaged over PcG-bound genomic regions (turquoise) and all CpG islands (red). (B) TI averaged over the TSSs of active genes (red), PcG targets (turquoise) and silent genes (blue) in ESCs. (C) The PcG-regulated *Hoxb* locus shows enrichment for H3.3 (top 4 tracks show H3.3 at each time point) but low TI (heatmap). The grey tracks show ChIP-seq for PcG protein Ring1b and the blue tracks RNA-seq. (D) The non-PcG-bound active gene *Hnrnpa3* shows greater H3.3 enrichment and higher TI than *Hoxb*. TI scale is the same as in *Figure 3*.

The following figure supplement is available for figure 4:

*Figure 4 continued on next page*

*Figure 4 continued*

**Figure supplement 1.** H3.3 enrichment at PcG targets in ESCs.

the most overrepresented genomic elements, similar to what was observed for the highest TI regions in ESCs. However, gene bodies were overrepresented in the NSC set but not in the set of top TI regions in ESCs (*Figure 5G*). This might reflect either unusually low turnover at gene bodies in ES cells, or high turnover at these sites in NSCs.

We conclude that H3.3 turnover is broadly similar in ESCs and NSCs with the exception of PcG-bound regions, which seem to stabilize during differentiation as evidenced by a loss of H3.3.

## Regions that change TI upon differentiation are often enhancers

Differences in H3.3 TI between pluripotent ESCs and lineage committed NSCs might identify regulatory regions important for differentiation. To explore this, we identified regions where TI changed upon differentiation. We compared TI between ESCs and NSCs for each 1 kb bin in the genome. Bins changing turnover were defined as those with a difference in TI between ESCs and NSCs in the top 2.5% with a p-value of ≤0.01. Using these criteria, 34,070 1 kb bins were identified which changed TI during differentiation (*Figure 6A* and *Supplementary file 3*) with more bins showing increased TI in NSCs compared to ESCs (19,870 bins compared to 14,200). Strikingly, regions showing higher TI in ESCs were enriched for the presence of an ESC enhancer (*Figure 6B*) and, similarly, regions with increased TI in NSCs were enriched for NSC enhancers (*Figure 6C*). Thus, enhancer regions showed high H3.3 turnover in a manner that correlated with cell type-specific enhancer activity. Interestingly, NSC enhancers were also somewhat overrepresented in regions with higher turnover in ESCs (*Figure 6B*). This could reflect roles for these enhancers in both ESCs and NSCs, or could reflect a 'poised' state for these enhancers (*Ferrari et al., 2014*; *Rada-Iglesias et al., 2011*). Regions with higher turnover in ESCs were also enriched for promoters and the 5' end of genes while rshowed a greater enrichment of gene bodies. In both sets, TTSs were overrepresented (*Figure 6B* and *Figure 6C*).

We observed TI differences in unannotated regions; 8732 of 34,070 differences, when regions overlapping UCSC genes were excluded. Such regions could represent unidentified elements with roles in cell identity and differentiation. This is supported by the fact that 1608 of these regions are located within 50 kb of a gene changing expression during differentiation (example in *Figure 6D*). When we examined only the 1000 largest turnover changes in unannotated regions, 212 regions were close to genes changing expression. In both instances the intersection between unannotated differences and these genes was statistically significant (p-value = 0.001), consistent with the possibility that these are previously unannotated regulatory regions.

We conclude that H3.3 turnover is regulated during differentiation and that the biggest changes in histone dynamics are found prominently in enhancer regions. Time-ChIP may be useful as a tool for picking out novel enhancers or other regulatory elements involved in cell fate changes.

## H3.3 turnover correlates with DNA accessibility at enhancers and TSSs

One attractive hypothesis is that histone turnover is directly related to accessibility of DNA sequences; regions of rapid turnover are expected to be more accessible. We wished to test this hypothesis by measuring the accessibility of DNA to enzymatic access genome-wide. DNase hypersensitivity, or the conceptually related technique ATAC-seq, detects regions of open chromatin (*Buenrostro et al., 2013*; *John et al., 2013*). We examined H3.3 turnover over previously annotated DNase hypersensitive sites in mouse ESCs (*Vierstra et al., 2014*) and found that TI was indeed high over these sites in ESCs (*Figure 7—figure supplement 1*). TI over ESC DNase sensitive regions was reduced in NSCs consistent with regulatory changes during differentiation (*Figure 7—figure supplement 1*). To expand upon this analysis, we used MNase, which is capable of measuring accessibility more broadly than these aforementioned technologies. It is established that nucleosomes have differential accessibility to MNase, with some nucleosomes liberated at low MNase and some requiring high MNase (*Knight et al., 2014*; *Xi et al., 2011*).

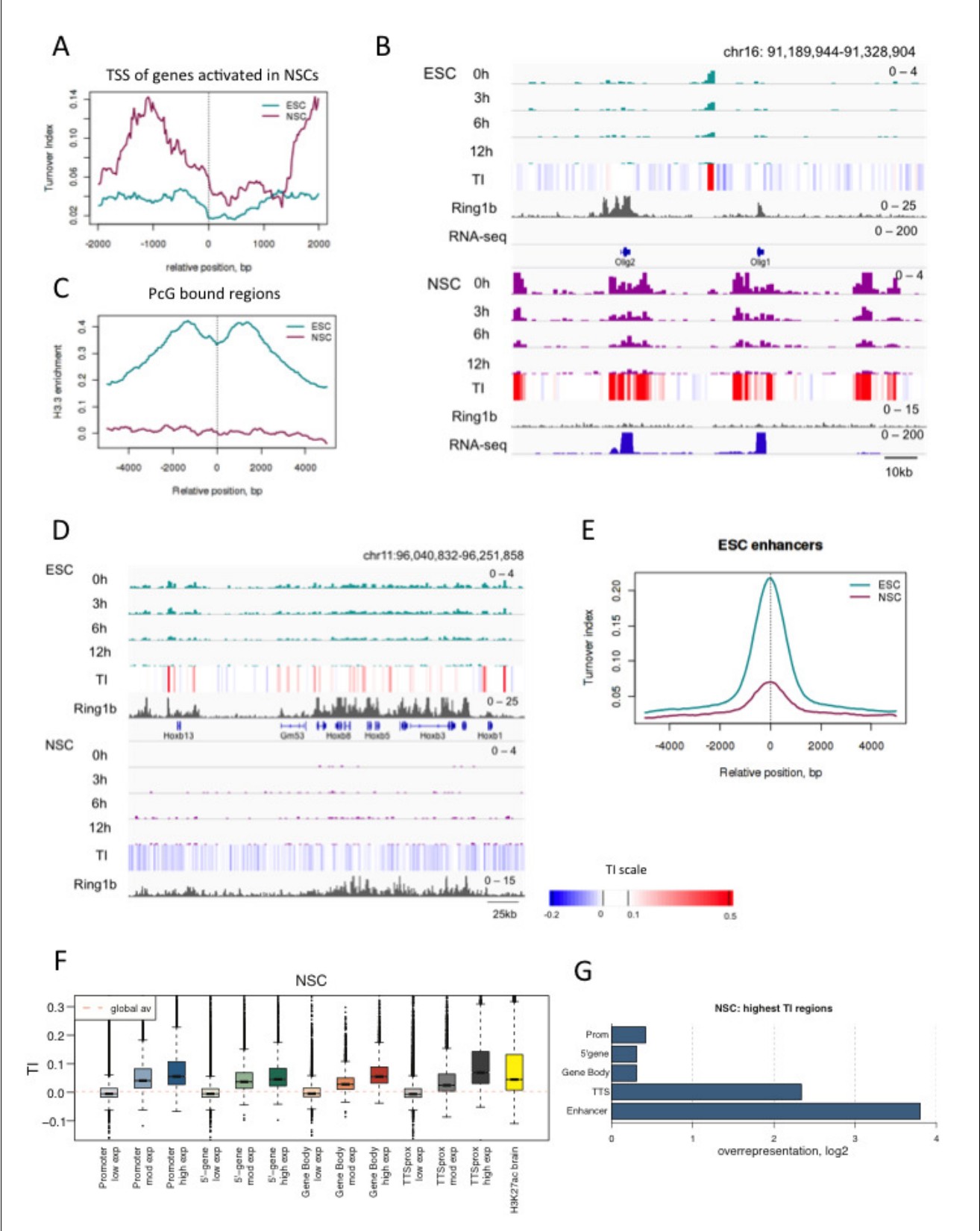

**Figure 5.** H3.3 turnover and distribution changes during neural differentiation of ESCs. (**A**) Average TI profile around TSSs of genes that lose PcG binding and increase gene expression upon differentiation to NSCs. (**B**) The region containing the *Olig1* and *Olig2* genes, which are activated during

*Figure 5 continued on next page*

*Figure 5 continued*
neural differentiation, shows a dramatic increase in H3.3 turnover when ESCs are differentiated to NSCs. The top tracks represent H3.3 (green) and TI (heatmap) in ESCs while the bottom tracks show H3.3 (purple) and TI in NSCs. The grey tracks show ChIP-seq for PcG protein Ring1b and the blue tracks RNA-seq. (C) Average H3.3 enrichment (0 hr sample) over PcG-bound regions in ESCs and NSCs. (D) *Hoxb* remains bound by PcG (Ring1b) in NSCs but H3.3 enrichment is lost. (E) Average TI at ESC-specific enhancers in ESCs (cyan) and NSCs (purple). (F) NSC TI distribution at annotated genomic elements – promoters, 5′ end of genes, gene bodies, TTSs and enhancers. Genes are grouped by NSC expression level and putative NS enhancers were defined using H3K27ac ChIP-seq data from Encode (E14.5 embryonic brain - ENCFF001XZR). (G) Representation of annotated genomic features in the 1000 highest TI regions in NSCs relative to their representation in the genome as a whole.
The following figure supplements are available for figure 5:

**Figure supplement 1.** Generation of H3.3_SNAP NSCs.
**Figure supplement 2.** Measuring turnover in H3.3_SNAP NSCs.

We have recently developed a metric that measures how nucleosomes respond to a titration of MNase. Nucleosomes that show decreased signal as MNase concentration decreases are deemed inaccessible (more MNase is needed to liberate them), while nucleosomes that show increased signal as MNase concentration decreases are deemed accessible (they are released at low MNase concentration and their associated DNA is over-digested at high concentrations; [*Mieczkowski et al., 2016*]) We therefore expanded upon the relationship between TI and DNase hypersensity to examine whether this new parameter and H3.3 turnover were correlated.

ESC and NSC chromatin was digested with a range of concentrations of MNase – 1U, 4U, 16U and 64U. We then calculated the 'MNase accessibility' (MACC) metric that reflects the change in signal at a particular locus in response to decreasing MNase concentration (*Figure 7—figure supplement 1*) (*Mieczkowski et al., 2016*). The metric is designed so that positive MACC values reflect regions that are easily digested by MNase and therefore are classed as 'accessible'. MNase titration was performed on two biological replicates of H3.3_SNAP ESC and NSC chromatin ('a' replicates). The same assay was also carried out on different ESC and NSC lines ('b' replicates) thus increasing the number of replicates we could assess and providing greater confidence in our findings.

We examined annotated genomic elements to see if turnover index correlated with DNA accessibility as assessed by MACC. At the TSSs of genes that change expression upon neural differentiation, TI and MACC were positively correlated in a cell-type specific manner (*Figure 7—figure supplement 2*). Increased turnover and accessibility at genes becoming activated during neural differentiation was evident at neuronal genes such as *Olig1* (*Figure 7—figure supplement 2*). In ESCs, at PcG-bound regions which are expected to be repressed, we observed slower turnover and also saw less accessibility (lower MACC) than enhancers and the TSSs of expressed genes (*Figure 7A*). These results are consistent with transcriptional repression at PcG-bound regions, although the presence of histone H3.3 at these regions implies that some replication-independent histone turnover occurs to deposit this histone.

We then examined DNA accessibility at ESC enhancers and super-enhancers as these regions show particularly high H3.3 turnover. We found that, in ESCs, TI and MACC correlated particularly well at these regions (*Figure 7B* and *Figure 7—figure supplement 2*). When the magnitude of TI and MACC at ESC enhancers was assessed, both had values well above the genome average in ESCs but not in NSCs, consistent with their regulatory role in pluripotent cells (*Figure 7C* and *Figure 7—figure supplement 2*). As well as being evident on a global scale, the relationship between TI and MACC at enhancers was also apparent at the level of individual loci. For example, the *Oct4/Pou5f1* super-enhancer shows high TI and high MACC in ESCs while both TI and MACC are decreased upon differentiation to NSCs (*Figure 7D*). These findings indicate that, at enhancers, increased turnover of nucleosomes might account for increased accessibility to the DNA sequence. This is consistent with increased transcription factor binding at these sites.

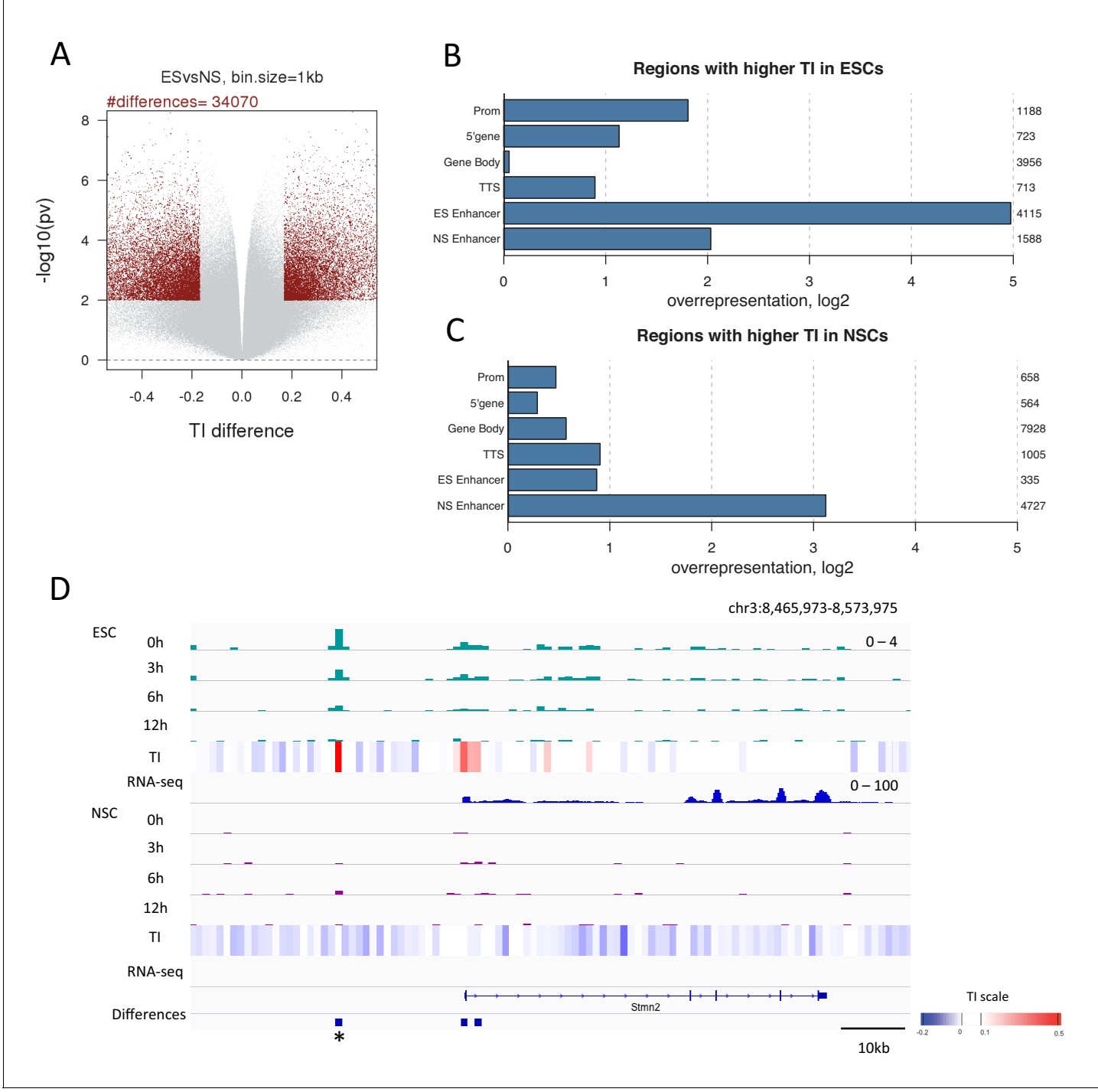

**Figure 6.** Changes in TI occurring during neural differentiation often occur at enhancers. (A) Volcano plot showing differences in TI between ESCs and NSCs. Regions with a TI difference in the top 2.5% with a p-value≤0.01 were selected as changing during differentiation (red dots). (B) Representation of annotated genomic features in regions showing higher TI in ESCs compared to NSCs (selected as described in A). The numbers to the right of the plot indicate the number of regions corresponding to each type of genomic feature. (C) Representation of annotated genomic features in regions showing higher TI in NSCs compared to ESCs (selected as described in A). (D) An unnanotated region on chromosome 3, which shows an increase in H3.3 and TI upon differentiation of ESCs to NSCs (asterisk) and an expression decrease in the nearby *Stmn2* gene. The heatmap shows average TI while differences in TI are represented by blue bars.

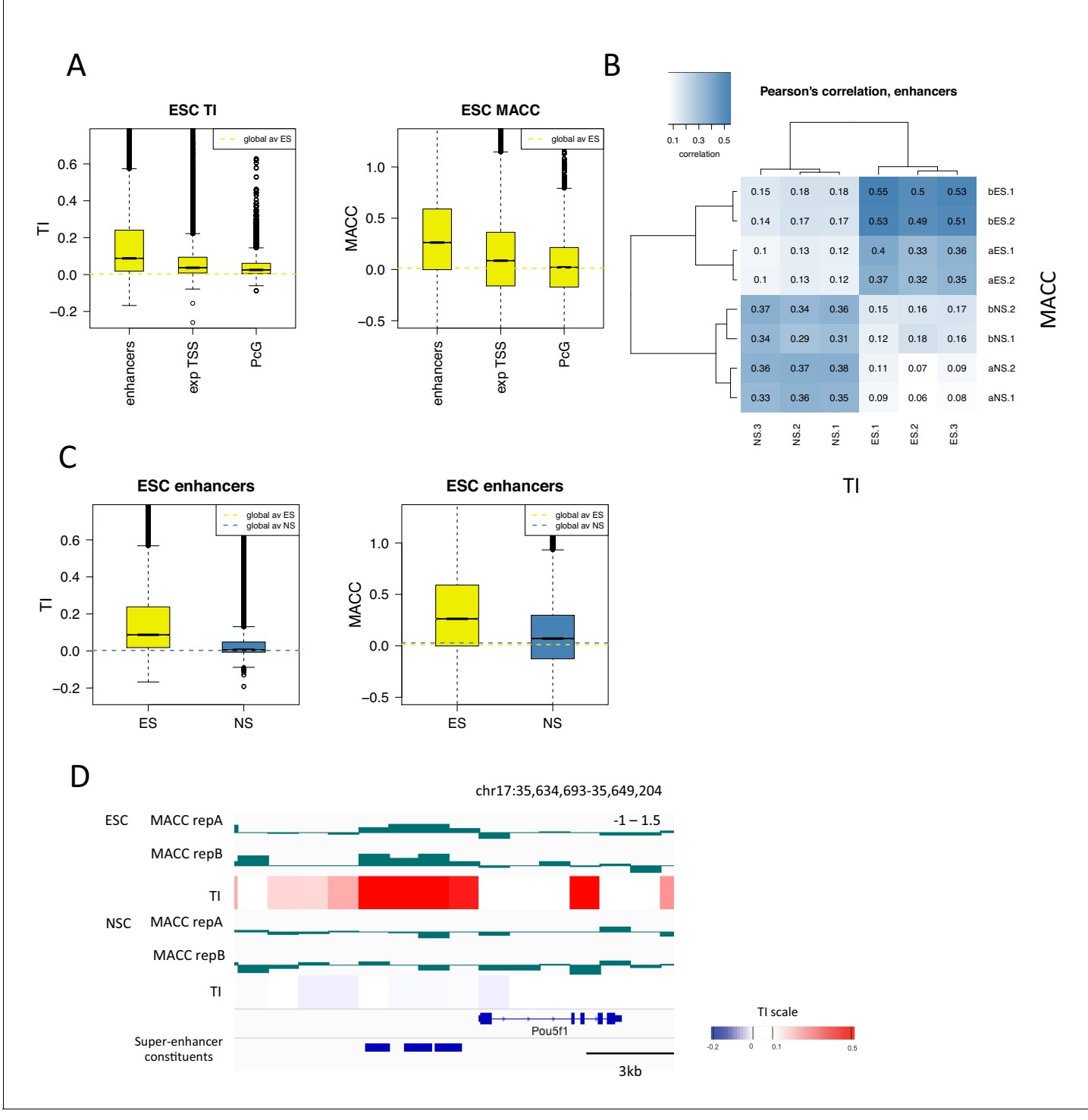

**Figure 7.** H3.3 turnover correlates with DNA accessibility at enhancers. (**A**) Boxplot of average ESC TI and MACC at ESC enhancers, the TSSs of expressed genes, and PcG-bound regions. (**B**) Pearson's correlation between DNA accessibility as measured by MNase titration (MACC) and TI at ESC enhancers. For MACC four replicates are shown for each cell type, two for ESC_ and NSC_H3.3_SNAP lines ('a' replicates) and two for separate ESC and NSC lines ('b' replicates). For TI three biological replicates are shown. (**C**) Boxplot of average TI and MACC values at ESC enhancers. (**D**) The *Oct4/Pou5f1* super-enhancer shows high MACC (green) and TI (heatmap) in ESCs but not in NSCs.

The following figure supplements are available for figure 7:

**Figure supplement 1.** Measuring DNA accessibility using MNase titration.

*Figure 7 continued on next page*

*Figure 7 continued*

**Figure supplement 2.** Comparing turnover to DNA accessibility measured by MNase titration.

## Discussion

Our unbiased assessment of H3.3 turnover identified enhancers as having highly dynamic nucleosomes. Notably, the clustered enhancers that make up super-enhancer domains were more rapidly turned over than conventional ESC enhancers. Measurement of H3.1 turnover showed the same phenomenon. These observations are consistent with the notion that nucleosomes must be disrupted in order to allow transcription factor binding and are consistent with increased DNA accessibility at enhancers. Notably, our data suggest that nucleosomes are frequently disrupted at enhancers regardless of which H3 variant they contain. Nucleosomes containing H3.3 predominate at these regions due to the high levels of nucleosome disruption but both H3.1 and H3.3 show rapid turnover. A variety of mechanisms might be used to generate high turnover in enhancers. Pioneer transcription factors such as Oct4 and Sox2 occupy these sets of ESC enhancers and super-enhancers and can bind partial motifs on the nucleosome surface (*Soufi et al., 2015*). These factors might contribute to nucleosome turnover, either directly through their binding or by recruiting remodeling complexes that facilitate turnover. Transcription through enhancers might also contribute (*Kim et al., 2010*). It is intriguing that super-enhancers show significantly higher turnover than enhancer elements. This could reflect a synergistic interaction between adjacent enhancer modules, or might indicate that the individual enhancer modules within super-enhancers each have intrinsically high turnover. In either case, the ability of turnover to identify these regions is striking and is likely to be related to their potency.

High histone turnover at enhancers was not limited to ESCs, but was also seen when we looked at lineage-committed NSCs, indicating that this property is seen both in pluripotent and in differentiated cell lineages. There were significant changes in where high turnover occurred in NSCs due to the activation of a distinct set of enhancers. In broad classifications, regions with rapid turnover were in general similar between ESCs and NSCs, except that gene bodies were overrepresented in high turnover regions in the differentiated cells but not in ESCs (*Figures 3* and *5*). Increased turnover in gene bodies might reflect widespread transcriptional activation and consequent histone turnover in differentiated cells. Alternatively, it might be that pluripotent cells have a plastic chromatin structure at the promoters of genes poised for activation in future lineages, increasing turnover relative to gene bodies at those locations.

Many unannotated regions showed differentiation-associated turnover changes, approximately one-quarter of turnover changes occurred outside of genes and enhancers. Many of these regions are likely to represent genomic elements with hitherto unknown roles in cell fate specification. In support of this, over 20% of the unannotated regions with the largest changes in turnover were located close to genes that changed expression during neural differentiation.

Nucleosome turnover might be related to accessibility of the underlying sequence as regions with rapid turnover are likely to have DNA sequences that are exposed for protein binding more frequently. To begin to analyze whether this is true, we measured accessibility genome wide using MNase, which is able to probe accessibility everywhere in the genome, and compared the resultant accessibility metric to turnover index. At ESC enhancers, both TI and MACC had high values and the two metrics were highly correlated. Similarly at the TSSs of expressed genes, TI and MACC had higher than average values while at repressed PcG targets both turnover and accessibility were reduced. These data thus cross validate these two distinct metrics and support the hypothesis that regions of high turnover correspond to regions of high DNA accessibility.

Our work and previous work are consistent in showing high incorporation rates for histone H3.3 and high turnover for that histone in enhancers and other regulatory regions (*Ha et al., 2014*; *Kraushaar et al., 2013*). We have extended these observations by also showing high turnover of H3.1 at enhancer regions and in demonstrating more rapid turnover of both H3.3 and H3.1 at super-enhancers compared to conventional enhancers. Notably, we also observed high H3.3 turnover around the transcription termination sites of expressed genes with our sets containing the (1000)

highest TI regions showing a large overrepresentation of TTSs. RNA Polymerase II occupancy or transcription itself could contribute to nucleosome displacement at these regions. Indeed, enrichment of RNA Pol II and nascent transcription at the 3' end of genes has been reported to be widespread (*Anamika et al., 2012*; *Core et al., 2008*; *Rahl et al., 2010*).

Time-ChIP allows examination of relative histone turnover in a genome-wide manner at high resolution, with changes in turnover evident at individual gene loci and regulatory elements. Labeling of SNAP-tagged histones for time-ChIP takes approximately one hour meaning that very rapid turnover might be missed and consequently turnover underestimated for some regions. Nevertheless, labeling time is relatively short compared to the cell cycle time of ESCs (12 hr) and NSCs (18 hr) and most H3.3 enriched loci are still detectable at the first time point as evidenced by the strong correlation between ChIP for H3.3_SNAP_HA and the H3.3 0 hr time-ChIP sample (*Figure 2—figure supplement 1*). We are therefore confident that we accurately quantify relative turnover for the vast majority of H3.3-enriched regions. In this study, H3.3 turnover was assessed because of its enrichment at regions important for gene regulation. In theory, however, time-ChIP can be applied to study the dynamics of any core histone or histone variant. We performed time-ChIP for H3.1 but a comprehensive assessment of its turnover, and that of other broadly distributed histones, is currently limited by the depth of sequencing required. Measuring nucleosome dynamics provides a distinct measure of chromatin characteristics that can help identify and functionally characterize regulatory elements.

## Materials and methods

### Experimental procedures

#### Generation and maintenance of H3_SNAP ESCs and NSCs

A p2lox plasmid containing H3.3 cDNA and a C-terminal SNAP_3xHA tag was introduced into A2lox.Cre ESCs (*Iacovino et al., 2011*) by nucleofection (Lonza, Switzerland). 24 hr prior to transfection 500 ng/ml doxycycline was added to induce Cre expression and facilitate integration of H3.3_SNAP_HA into the *hprt* locus. 24 hr after transfection selection was initiated with 300 ug/ml G418 and after 7–9 days colonies were picked and screened for presence of H3.3_SNAP_HA. The resulting A2lox.H3.3_SNAP ESCs were maintained on mitomycin-C inactivated embryonic fibroblast feeders in DMEM supplemented with 15% fetal bovine serum (Hyclone) and 1000 U/ml of leukemia inhibitory factor (EMD Millipore). H3.1_SNAP ESCs were generated in the same manner using a p2lox plasmid containing H3.1 cDNA and a C-terminal SNAP_3xHA tag.

H3.3_SNAP NSCs were generated by injecting H3.3_SNAP ESCs into blastocysts and harvesting the resulting embryos at E13.5. Neurospheres isolated from embryonic brain were grown in suspension for 1–2 passages and then plated on poly-l-ornithine coated plates and grown as a monolayer in DMEM:F12 containing B-27 and N-2 supplements (Gibco) and EGF+FGF-2 (10 ng/ml). NSC lines were screened for chimerism based on puromycin resistance (which is present in the *Rosa26* locus of H3.3_SNAP cells). The most highly chimeric lines were selected with puromycin to generate pure populations of H3.3_SNAP NSCs and expression of H3.3_SNAP_HA was tested after 24 hr doxycycline induction by HA staining and flow cytometry. The best H3.3_SNAP NSC line (line 3_1) was then chosen for further experiments.

#### Time-ChIP: Biotin labeling and histone pulldown

Prior to labeling, H3.3_SNAP or H3.1_SNAP was induced with 500 ng/ml doxycycline for four cell doubling times (48 hr for ESCs and 72 hr for NSCs). For labeling, $0.5–2 \times 10^8$ cells were incubated in 50 µl complete medium supplemented with 10 µM CP-Biotin (NEB) for 40 min at 37°C then washed twice with 10 ml media. Cells were then resuspended in 10 ml complete medium and incubated for 20 min at 37°C to remove any unbound CP-biotin. After a final wash, one-fifth of the labeled cells were harvested as the 0 hr time point and the rest plated for harvest at 3 hr, 6 hr or 12 hr.

Nuclei were extracted from each sample, counted, and equal numbers of cells resuspended in 500 µl 20 mM Hepes pH7.7, 20 mM KCl, 0.5 mM EDTA, 300 mM NaCl, 1 mM DTT containing complete protease inhibitors (Roche). Nuclei were digested with 120–140U MNase (Worthington) in the presence of 3 mM $CaCl_2$ for 20 min at 25°C. Reactions were stopped by the addition of EDTA/EGTA and chromatin was solubilized by the addition of 0.05% NP-40. The MNase digested chromatin was

centrifuged at 12,000 g for 5 min. 400 µl of the resulting supernatant was added to 15 µl streptavidin magnetic beads (Pierce) which had been pre-blocked with 50 mg BSA and 200 µg/ml yeast tRNA and incubated for 1 hr at 4°C. Beads were washed 3 × 5 min with 20 mM Hepes pH7.7, 20 mM KCl, 500 M NaCl, 3 mM CaCl2, 5 mM EGTA, 0.5% NP40, 1 mM DTT containing complete protease inhibitors (Roche). DNA was eluted with 250 µl 10 mM Tris pH8, 100 mM NaCl, 10 mM EDTA, 0.5% SDS and incubated for 10 min at at 30°C in the presence of RNase A (DNase-free, Roche). DNA was purified by phenol-chloroform and chloroform extraction followed by column purification (PCR purification kit, Qiagen). DNA was quantified using picogreen (Molecular Probes).

Three biological replicates were performed for time-ChIP experiments based on what is standard for ChIP-seq experiments. For each replicate, cells were cultured at different times. They were then labeled, harvested at various time-points post-labeling. Pulldown was performed separately for each replicate.

## Chromatin immunoprecipitation (ChIP)
### Native ChIP for H3.3_HA
H3.3_SNAP_HA was induced with doxycycline for 48 hr, approximately 0.5–1 × 10$^7$ cells were harvested, and washed in cold PBS. Nuclei were isolated and chromatin was prepared and fragmented by MNase as described above. Chromatin was added to 15 µl protein A dynabeads (Life Technologies) that had been preincubated with anti-HA (Abcam, ab9110) and incubated overnight at 4°C. Washing, elution and DNA purification was carried out as described above.

### Cross-linked ChIP
Cells were cross-linked in 1% formaldehyde at room temperature and ChIP carried out as described in Illingworth et al. (*Illingworth et al., 2015*) with minor modifications. For Ring1b ChIP cells were crosslinked with 2 µM EGS (Thermoscientific) at room temperature for 40 min prior to formaldehyde fixation. Antibodies used were as follows anti-H3K27me3 (Millipore, 07–449), anti-Ring1b (Bethyl, A302-869A), anti-Cbx2 (Santa Cruz, sc-19297), anti-Ezh2 (Cell Signaling 5246S).

## Transcriptional analysis
RNA was isolated using TRIZOL and Purelink RNA mini kit (Ambion – Life Technologies). For RT-qPCR, RNA was reverse transcribed using SuperScript III Reverse Transcriptase (Life Technologies) and the resulting cDNA analyzed by qPCR using iTaq Universal SYBR Green Supermix (Bio-rad) on an ABI 7500 fast qPCR machine. Primer sequences are listed in *Supplementary file 6*. For RNA-seq experiments, rRNA was depleted using Ribo-Zero Gold kit (Illumina). First and second strand cDNA synthesis were performed using the TruSeq RNA library prep kit (Illumina) according to manufacturer's instructions. Library preparation was carried out as described below.

## Western blotting
Protein extracts were denatured at 95°C for 5 min in 3x reducing loading buffer (NEB), separated by SDS-PAGE using a 4–12% NuPAGE Bis-Tris gel (Life Technologies), then transferred to PVDF membrane (Bio-Rad, for immunofluoresence). Membranes were blocked, incubated with antibodies and washed according to standard protocols. Primary antibodies used were anti-HA.11 (Covance, MMS-101P), anti-H3 (Cell signaling, 4499L), anti-H3.3 (Millipore, 09–838), anti-H3K4me3 (Millipore, 07–473), anti-H3K27me3 (Active Motif, 39155) and anti-TBP (Abcam, ab818). Secondary antibodies were LI-COR fluorescent secondary antibodies or HRP-conjugated. Membranes were imaged on the LI-COR Odyssey Sa or using Pierce ECL (Thermo Scientific) and film.

## Immunofluorescence
Cells were grown on coverslips and immunofluorescence was performed according to standard protocols. Primary antibodies used were anti-Sox2 (Millipore) and anti-Nestin (DSHB, rat-401). Secondary antibodies were anti-rabbit-IgG-Cy3 and anti-mouse-IgG-488. Coverslips were mounted in Vectashield with DAPI.

## MNase titration

MNase titration was carried out as described in Mieczkowski et al. In brief, cells were harvested, cross-linked in 1% formaldehyde at room temperature and nuclei isolated. Digestions were performed on $2.5 \times 10^5$ nuclei per reaction using 64U, 16U, 4U or 1U MNase (Worthington) in 10 mM Tris pH 7.4, 15 mM NaCl, 60 mM KCl, 1 mM CaCl2 for 10–15 min at 25°C. Reactions were stopped with EDTA/EGTA and 0.5% SDS and 125 mM NaCl was added to the samples. RNase A and proteinase K treatment along with cross-link reversal at 65°C were performed. DNA was purified by phenol-chloroform extraction and column purification.

## Illumina HiSeq library preparation and sequencing

Sequencing libraries were prepared as described in Bowman et al. (*Bowman et al., 2013*). For time-ChIP libraries, the same amount of spike-in control DNA was added to each library. The amount of spike-in DNA added to each library was 10,000 times less than the amount of DNA recovered from the 0 hr time-point. The spike-in control consisted of synthetic 150 bp fragments of DNA with GC content of 30–70%. These sequences can be found in the Supplementary Material (*Supplementary file 5*). Paired-end 50 sequencing was carried out in an Illumina HiSeq 2500 according to manufacturer's instructions. For time-ChIP libraries, four samples were barcoded and run in one sequencing lane, yielding approximately 40–50 million reads for each sample.

## Bioinformatics and statistical analysis

### Sequencing alignment and quality control

Sequenced 50-bp paired-end tags were mapped to the mouse (mm 9) genome using the Bowtie aligner v. 0.12.9 (*Langmead et al., 2009*). For time-ChIP samples, where tags had 5 or more reportable alignments, one was reported at random. For MACC calculations, only uniquely mapped tags were considered. In addition, only tags with no more than two mismatches were retained. Genomic positions with the numbers of mapped tags above the significance threshold of z-score = 7 were identified as anomalous, and the tags mapped to such positions were discarded. Coordinates of genes were from the RefSeq mm9 gene set. The TSS profiles were calculated and plotted as described previously (*Tolstorukov et al., 2012*).

### Spike-in control

For each time point the number of reads aligning to the spike-in control was compared to the number of reads aligning to the mouse genome. The greater the number of spike-in reads relative to mouse genomic reads, the smaller the quantity of genomic DNA sequenced from the time point sample. We used this relationship to calculate the relative number of 'sequencing reads' for each time point relative to the 0 hr time point (*Figure 1—figure supplement 3*).

### Time-ChIP: TI calculation

The genome was divided into 1 kb bins and, for each sample, the number of tags per bin was calculated as reads per million to normalize for library size. Data for each time point was then input subtracted. For each 1 kb bin, TI was calculated by fitting the number of reads for each time point to a simple linear regression model. A Kendall rank correlation test was used to assess how well points fit the linear model. The slope of the line fitted by linear regression was weighted by multiplying by 1-p-value from the Kendall test. Such weighting was introduced to reduce the TI values for the bins where the statistical significance of the fit was low, and thus it ensures a conservative estimation of TI (*Abubucker et al., 2012*). Finally, the sign of the slope was reversed such that positive TI values represented high turnover. As data for individual biological replicates correlated well (*Figure 2—figure supplement 1*), the mean of these replicates was used for further analysis. In *Figure 3—figure supplement 1* a 300 bp bin size was used for calculating TI.

### Identification of top TI regions

To identify regions with the highest H3.3 turnover a sliding window calculation was performed that calculated the mean TI over a window size of three 1 kb bins. This allowed us to identify high TI bins whose neighboring bins also had high turnover. Regions corresponding to repeats were manually removed and the 1000 bins with the highest TI value were chosen as the set of highest TI regions for

the cell type in question. An FDR threshold was calculated for these regions by taking the lowest TI value in this set and seeing how frequently it occurred in 1000 randomly selected 1 kb bins. This random selection was performed 100 times and for each test; FDR = # bins with TI value above threshold/# total bins. The highest value from these 100 tests was chosen as the final FDR.

### Detecting differences in TI between ESCs and NSCs

TI difference for each 1 kb bin was calculated (mean ES TI – mean NS TI) and p-values were computed using the empirical Bayes method (limma package in R) on TI data from each of the three replicates. Bins with a TI difference in the 0.975 quantile with p-values $\leq$ 0.01 were selected as changing TI during differentiation.

### Definition of unannotated regions and overlap with genes

Regions were defined as occurring in an unannotated region if they did not overlap a gene or an enhancer element. To ensure elements were truly unannotated we extended our gene set to include all UCSC mm9 genes and predicted genes (52,199 genes). To determine whether unannotated regions of interest overlapped particular gene sets in a statistically significant manner we used GAT (http://gat.readthedocs.org/en/latest/usage.html) (*Heger et al., 2013*).

### MACC calculation

MACC was calculated as described in Mieczkowski et al.

### ChIP-seq analysis

To determine regions of ChIP-seq tag enrichment (peaks), we analyzed tag counts in a 1 kb window over the chromosome length with the step of 200 bp. Statistical significance of enrichment of ChIP vs input was estimated using negative binomial distribution, with the estimate of the mean based on the tag counts in input, and the size parameters selected based on manual inspection of resulting peak calls. Regions of significant enrichment were generated by merging adjacent significantly enriched windows separated by 1 kb or less. In ESCs, PcG targets were defined as having overlapping peaks of Ring1b Cbx2, Cbx7, Ezh2 and H3K27me3. For NSCs, PcG targets were defined as having overlapping peaks of Ring1b and H3K27me3.

### Gene expression

Expression data for TSS plots in *Figure 2* and *Figure 4* was taken from West et al. (*West et al., 2014*). Expression data in *Figures 3*, *5* and *7* is from RNA-seq data performed on J-1 ESCs and NSCs differentiated in vitro from these cells. RNA-seq tags were aligned to the mouse genome (mm9) using the Tophat software package with default parameters (*Trapnell et al., 2009*). FPKM values and differential gene expression between ESC and NSC were calculated using Cufflinks (*Trapnell et al., 2010*) and Cuffdiff (*Trapnell et al., 2013*) packages respectively. Genes changing expression between ESCs and NSCs were defined as those showing at least a three-fold (for comparison with unannotated regions changing turnover during differentiation) or a four-fold change (MACC analysis) in FPKM value.

### Data availability

High throughput sequencing data has been deposited in GEO and is accessible using the following links.

Time-ChIP: GSE78876
http://www.ncbi.nlm.nih.gov/geo/query/acc.cgi?acc=GSE78876
ChIP-seq: GSE78899
http://www.ncbi.nlm.nih.gov/geo/query/acc.cgi?acc=GSE78899
MNase titration and RNA-seq: GSE78984
http://www.ncbi.nlm.nih.gov/geo/query/acc.cgi?acc=GSE78984

## Acknowledgements

We thank Ivan Correa (NEB) for the gift of the CP-biotin substrate, Michael Kyba for supplying the A2lox.Cre ESCs and Sihem Cheloufi for assistance isolating embryonic NSCs. MGR and LETJ are supported by the Fundacão para a Ciência e a Tecnologia (FCT), EMBO and the ERC. REK is supported by the NIH.

## Additional information

### Funding

| Funder | Grant reference number | Author |
| --- | --- | --- |
| National Institutes of Health | NIH R01 GM48405 | Robert E Kingston |
| Fundação para a Ciência e a Tecnologia | BIA-PRO/100537/2008 | Lars ET Jansen |
| Fundação para a Ciência e a Tecnologia | SFRH/BD/33567/2008 | Mariluz Gómez-Rodríguez |
| European Molecular Biology Organization | installation grant 1818 | Lars ET Jansen |
| European Research Council | ERC-2013-CoG-615638 | Lars ET Jansen |

The funders had no role in study design, data collection and interpretation, or the decision to submit the work for publication.

### Author contributions

AMD, Conception and design, Acquisition of data, Analysis and interpretation of data, Drafting or revising the article; MGm-Rǵuez, LETJ, Conception and design, Analysis and interpretation of data, Drafting or revising the article, Contributed unpublished essential data or reagents; JM, MYT, RIS, REK, Conception and design, Analysis and interpretation of data, Drafting or revising the article; SK, Conception and design, Acquisition of data, Drafting or revising the article

### Author ORCIDs

Robert E Kingston, http://orcid.org/0000-0003-3628-4335

## Additional files

### Supplementary files

• Supplementary file 1. Top 1000 regions with highest turnover in ESCs.

• Supplementary file 2. Top 1000 regions with highest turnover in NSCs.

• Supplementary file 3. Differences in turnover between ESCs and NSCs.

• Supplementary file 4. External datasets used.

• Supplementary file 5. Spike-in controls.

• Supplementary file 6. Primer sequences.

### Major datasets

The following datasets were generated:

| Author(s) | Year | Dataset title | Dataset URL | Database, license, and accessibility information |
|---|---|---|---|---|
| Deaton A, Gomez-Rodriguez M, Mieczkowski J, Tolstorukov MY, Kundu S, Sadreyev R, Jansen LE, Kingston RE | 2016 | Time-ChIP in ESCs and NSCs | http://www.ncbi.nlm.nih.gov/geo/query/acc.cgi?acc=GSE78876 | Publicly available at the NCBI Gene Expression Omnibus (Accession no: GSE78876) |
| Deaton A, Gomez-Rodriguez M, Mieczkowski J, Tolstorukov MY, Kundu S, Sadreyev R, Jansen LE, Kingston RE | 2016 | ChIP-seq for Polycomb group proteins and H3K27me3 in ESCs and NSCs | http://www.ncbi.nlm.nih.gov/geo/query/acc.cgi?acc=GSE78899 | Publicly available at the NCBI Gene Expression Omnibus (Accession no: GSE78899) |

The following previously published datasets were used:

| Author(s) | Year | Dataset title | Dataset URL | Database, license, and accessibility information |
|---|---|---|---|---|
| Mieczkowski J, Cook A, Bowman SK, Mueller B, Alver BH, Kundu S, Deaton AM, Johnson J, Larschan E, Park PJ, Kingston RE, Tolstorukov M | 2016 | MNase titration reveals differences between nucleosome occupancy and chromatin accessibility | http://www.ncbi.nlm.nih.gov/geo/query/acc.cgi?acc=GSE78984 | Publicly available at the NCBI Gene Expression Omnibus (Accession no: GSE78984) |
| Ren | 2012 | H3K27ac ChIP-seq on 14.5 day embryonic mouse whole brain | http://www.ncbi.nlm.nih.gov/geo/query/acc.cgi?acc=GSM1000094 | Publicly available at the NCBI Gene Expression Omnibus (Accession no: GSM1000094) |
| Stamatoyannopoulos | 2013 | DNase on embryonic mouse stem cell line E14 | http://www.ncbi.nlm.nih.gov/geo/query/acc.cgi?acc=GSM1014154 | Publicly available at the NCBI Gene Expression Omnibus (Accession no: GSM1014154) |
| Hardison | 2014 | PSU mouse ES-E14 H3K9me3 ChIP-seq | https://www.encodeproject.org/experiments/ENCSR857MYS/ | Publicly available at the Encyclopedia of DNA Elements (Accession no: ENCSR000CMW) |
| West JA, Cook A, Alver BH, Stadtfeld M, Deaton AM, Hochedlinger K, Park PJ, Tolstorukov MY, Kingston RE | 2014 | Nucleosome occupancy changes in mammalian cell differentiation and reprogramming | http://www.ncbi.nlm.nih.gov/geo/query/acc.cgi?acc=GSE59064 | Publicly available at the NCBI Gene Expression Omnibus (Accession no: GSE59064) |

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
