## [Decision Letter]

[Editors’ note: this article was originally rejected after discussions between the reviewers, but the authors were invited to resubmit after an appeal against the decision.]

Thank you for submitting your work entitled "Enhancer regions show high histone H3.3 turnover that changes during differentiation" for consideration by *eLife*. Your article has been favorably evaluated by Jessica Tyler (Senior editor) and three reviewers, one of whom is a member of our Board of Reviewing Editors. Our decision has been reached after consultation between the reviewers. Based on these discussions and the individual reviews below, we regret to inform you that your work will not be considered further for publication in *eLife*.

The reviewers all applauded the new methods developed in the manuscript and the promise that these methods will lead to advances in the field. However, it was felt that such advances and novel discoveries were not contained in the manuscript submitted. Most of the results confirm observations previously made with other approaches. It was suggested that the current manuscript would be more appropriate for a high impact methods journal. The reviews also contain several comments and technical points that we help you will find helpful.

Reviewer #1:

This manuscript uses a novel technique to measure histone H3.3 turnover to generate a turnover index. The experiments are carefully done, well presented and well written. The results are largely predictable (e.g. high turnover at accessible act sites and low turnover at less accessible inactive sites).

1) While the method is commendable my sense is that to have a high enough impact for *eLife* the manuscript really needs to emphasize novel findings from this method that we did not know already.

2) Why study only the replacement variant of H3. Comparison to H3.1 would have increased the impact of the work. It might also allow some measurements on the rate of turnover during the cell cycle.

3) Other H3 tags have been shown to interfere with modification of the histone. This study needs some general controls for the stability of the nucleosomes containing the tagged histone and for the presence of histone modifications on the tagged vs. endogenous histone.

Reviewer #2:

In the manuscript entitled "Enhancer regions show high histone H3.3 turnover that changes during differentiation" Deaton et al. developed a new approach to study variant histone H3.3 turnover and performed a thorough systematic investigation in mouse embryonic stem cells (mESCs) and neural precursors. They correlate such turnover with ESC promoters, with gene expression and with other metrics that allow regulatory region identification (i.e., H3K27ac and enhancers). The authors also employed their newly developed method called time-ChIP to assess histone turnover genome-wide after differentiation of mESCs to neural stem cells (NSCs), albeit differentiation occurred in mouse embryo brains. The authors found high histone turnover in the regions that are involved in gene regulation particularly, enhancers and super-enhancers. The turnover is quantitated by a turnover index (TI) in those regions. The results were further corroborated by MNase accessibility metric (MACC), a measure of chromatin/DNA accessibility. The overall approach is solid but we did not see a significant advantage over previously published approaches, i.e., the TET-OFF approach to H3.3 turnover by Zhao's lab. Also, the authors claim that their dual approach would be useful in identifying novel regulatory regions, but they did not pursue this avenue of investigation in a tangible way.

Figure 1 described the general approach to histone H3.3 tagging and thence turnover but lacked good controls; Figure 1—figure supplement 1 was somewhat ambiguous. This needs to be a crystal clear figure with obvious controls – it's not. The authors point out that much of the data of Figure 2 are as expected based on previous studies of H3.3 and known histone turnover in active regions vs. less turnover in inactive. The conclusions are compatible with the work from Henikoff, Allis and specifically Zhao's 2014 Epigenetics Chromatin paper on H3.3 turnover in mESC using a TET-OFF strategy. The study does not consider the presence of fragile nucleosomes. Figure 3 shows turnover correlates with gene activity, promoters and enhancers, particularly super-enhancers. The analysis here was a bit deeper than in the Zhou paper but also with a slightly different emphasis. The authors then concentrated on polycomb-repressed genes in Figure 4 and show a lower turnover, again, as expected. Figure 5 revealed a whole new set of genes that are activated upon neural differentiation, which had lost polycomb (PcG) binding, but again as expected these data showed a dramatic increase in H3.3 enrichment and histone turnover, whereas regions that are still bound by PcG in NSCs, were no longer enriched in H3.3 as they were in ESCs. Interestingly, gene bodies showed high turnover in NSC vs ESC. Remarkably, the NSC lines derived from embryos had rather rapid cell cycles vs. some generated in culture. Figure 7 is an ESC-NSC comparison but we found nothing striking.

Strengths: This study offers two potentially useful tools to study nucleosome dynamics: first, to quantify the histone turnover rate (TI) and second, the MNase accessibility metric (MACC); the latter study appears to also be submitted elsewhere. The results are clean and the data presented in this study are largely convincing. The approach may indeed provide insight into identification of novel regulatory regions but the authors need to prove it.

Weaknesses: This study does not break new conceptual ground or add significantly to our present understanding of nucleosome dynamics linked with transcription and silencing. It has long been known from a number of studies that gene expression positively correlates with the occupancy of H3.3 in the regulatory regions (especially promoters and enhancers) and generally associates with high nucleosome dynamics.

Reviewer #3:

This paper describes the application of a novel technique (time-ChIP) involving an induced histone variant coupled to a SNAP tag. The system is used in this paper to determine the genome-wide histone turnover of the H3.3 variant in a developmental context (ES to neuronal ES). The authors conclude that regions of the genome experiencing active regulation (promoters, enhancers, and super-enhancers) experience faster histone turnover relative to silenced or inactive regions. Additionally, the authors describe a relationship between polycomb repressive complex and the stabilization of chromatin during neuronal differentiation. Finally, the authors propose this technique as a potential method of identifying previously uncharacterized active regulatory regions.

At first glance, the study seems interesting and relevant to the field of nucleosome regulation. However, there were several analytical steps that lacked appropriate details which significantly impaired the full interpretation of the analysis and conclusions. It is difficult to recommend for publication without additional details of the methodology to make an informed decision.

A more detailed description of how the spike-in control was used is critical. How was this spike-in used to normalize or control the data?

The authors state that they achieved greater DNA recovery the closer to the pulse they collected chromatin which is expected and serves as a good internal control. However, if the genome-wide normalization is based on the relative chromatin enrichment, the TI calculation could merely be mirroring the increased DNA recovery and not, on any sites, actual turnover. Further details on how the data was specifically normalized would clarify this concern. Additionally, it would be important to know the robustness of this TI calculation given alternative normalization approaches. For example, the authors show there is minimal turnover at silent genes. If this is expected to be a relative constant, how do their findings hold up if normalized to turnover at silent genes?

What is the rationale for the use of 1kb windows for the calculation of the TI index? This time-ChIP data is presumably at nucleosome resolution since it is MNase-based. It would be of far greater value to see the extent of histone turnover extending into genes rather than what appears to be arbitrary 1 kb blocks. If there is a reason why the data is kept at 1kb blocks, the authors should also explain the limits of the spatial resolution of their data.

The use of linear regression slopes as the TI index raises concerns that the authors are assuming a linear relationship in histone turnover. What about regions that experience little turnover for long periods followed by spurts of high turnover? Would the TI metric pull that data out or would it be removed from consideration because of the weighting schema used to punish poorly fit regression models? Again this speaks to the concerns regarding the robustness of the TI metric.

How is the MACC calculated? There is insufficient information for the nature of this calculation, which is concerning since an entire section of the paper is based on this calculation. A reference to a submitted paper that was not provided is insufficient to judge the nature of the analysis and findings.

[Editors’ note: what now follows is the decision letter after the authors submitted for further consideration.]

Thank you for resubmitting your work entitled "Enhancer regions show high histone H3.3 turnover that changes during differentiation" for further consideration at *eLife*. Your revised article has been favorably evaluated by Jessica Tyler (Senior editor), a Reviewing editor, and two reviewers.

The manuscript has been improved but there are some remaining issues that need to be addressed before acceptance, as outlined below:

This is a highly interesting study and certainly contributes to our understanding of nucleosome dynamics at enhancers and super-enhancers. However, it is mainly a techniques paper that states that there is high turnover at enhancers/super-enhancers and extends the work of Zhao while adding a large amount of new detail. We point out that Zhao's 2014 Epigenetics paper used the TET-OFF strategy referred to by the author in his rebuttal but Zhao also has a very nice Genome Bio paper, PMC3983652, that was cited by the author and which incorporates many and earlier time points. Although performed in MEFs, that study also shows H3.3 at high resolution and demonstrate high H3.3 turnover at enhancers and low at H3K27me3 rich regions using linear regression models in a TET-ON H3.3 system. The author's new strategy is probably more accurate versus attempting to measure turnover by regression.

1) While the Discussion does in fact mention that Zhao came to the same conclusion on enhancers and cites the right paper, it comes too late in the manuscript. Hence, the Abstract makes an inappropriate novelty claim regarding high H3.3 turnover at regions known to be involved in gene regulation. Since Zhao has already shown this then they should acknowledge it in the Abstract and Introduction, i.e., "As previously shown, we found high H3.3 turnover at active regulatory regions including enhancers. However, we also find that even higher turnover occurs at super-enhancers[…]".

2) The authors have also satisfied the technical concerns that were raised in the previous review.

3) The revised version now included the requested statistical information.

---

## [Author Response]

[Editors’ note: the author responses to the first round of peer review follow.]

Reviewer #1:

1) While the method is commendable my sense is that to have a high enough impact for eLife the manuscript really needs to emphasize novel findings from this method that we did not know already.

We respectfully dispute the statement concerning ‘novel findings’; while many in the chromatin field take as a given that regulatory regions such as enhancers will have high turnover, there is scant data on this point, and the data that exists is not particularly compelling. The previous work uses protein induction strategies that have time delays, lessening precision. In addition, the difference between super-enhancers and enhancers is novel, as is the extent to which turnover is increased over gene bodies upon differentiation. We have examined turnover during differentiation, which has not to our knowledge been previously studied, and find that enhancers show the most frequent changes in turnover. Enhancers are central and highly studied, these are key data on a key topic.

2) Why study only the replacement variant of H3. Comparison to H3.1 would have increased the impact of the work. It might also allow some measurements on the rate of turnover during the cell cycle.

We have the H3.1 data on enhancers, and had not included it in the initial version, as we do not have read depth to allow assessment of any other feature of the genome. (The reason H3.1 is difficult to study intensively right now is necessary read depth and attendant cost.) The data are noisy due to read depth even at enhancers. We now include these data (Figure 3—figure supplement 2) and it buttresses the H3.3 data.

3) Other H3 tags have been shown to interfere with modification of the histone. This study needs some general controls for the stability of the nucleosomes containing the tagged histone and for the presence of histone modifications on the tagged vs. endogenous histone.

Looking at modifications on the tagged histone is tough due to the low expression level, however we can see K4me3 on H3.3 and K27me3 on H3.1 and have included these data (Figure 1—figure supplement 2). All of our data speaks to the stability of the tagged histone on chromatin (where we are using it) as we find turnover numbers that span hours and that are concordant with previous estimates. For example, Kraushaar et al. find H3.3 incorporation to occur in a 3-24 hour time frame. This has been added to the text (Results section).

Reviewer #2:

Weaknesses: This study does not break new conceptual ground or add significantly to our present understanding of nucleosome dynamics linked with transcription and silencing. It has long been known from a number of studies that gene expression positively correlates with the occupancy of H3.3 in the regulatory regions (especially promoters and enhancers) and generally associates with high nucleosome dynamics.

See above for the rebuttal on novelty; again the data are scant and no one has rank ordered turnover in active genes vs. enhancers vs. super-enhancers previously and these are high impact issues.

Reviewer #3:

At first glance, the study seems interesting and relevant to the field of nucleosome regulation. However, there were several analytical steps that lacked appropriate details which significantly impaired the full interpretation of the analysis and conclusions. It is difficult to recommend for publication without additional details of the methodology to make an informed decision.

A more detailed description of how the spike-in control was used is critical. How was this spike-in used to normalize or control the data?

The spike-in is a control to verify that the proportion of sequencing reads resulting from pull-down of SNAP-tagged histones was consistent with the experimental data for each time point, one cannot normalize due to the nature of the experiment. We apologize for not having made this clear, and have expanded upon this in the text (subsection “Time-ChIP reports histone H3.3 turnover genome-wide”, second paragraph and Materials and methods).

The authors state that they achieved greater DNA recovery the closer to the pulse they collected chromatin which is expected and serves as a good internal control. However, if the genome-wide normalization is based on the relative chromatin enrichment, the TI calculation could merely be mirroring the increased DNA recovery and not, on any sites, actual turnover. Further details on how the data was specifically normalized would clarify this concern.

Although we used a spike in control (as described above) we did not perform any normalization of the sequencing data. Rather, we look at the relative number of reads for different genomic regions at each time point when assessing turnover. Thus these data are internally compared to each other; the turnover rates reflect that internal comparison. For example, high turnover regions such as enhancers will have fewer reads in later time points compared to regions with slower turnover and the TI calculation reflects this. We apologize for not making this clear initially and have expanded upon this in the text (subsection “Time-ChIP reports histone H3.3 turnover genome-wide”, fifth paragraph).

Additionally, it would be important to know the robustness of this TI calculation given alternative normalization approaches. For example, the authors show there is minimal turnover at silent genes. If this is expected to be a relative constant, how do their findings hold up if normalized to turnover at silent genes?

As discussed above the read counts are not normalized and we compare relative turnover rates between different genomic regions. By its nature, the TI metric is internally “normalized” across the genome as we look at turnover at silent sites as well as active regions, and the metric provides a direct comparison between these rates. The silenced regions, such as those occupied by Polycomb, have lower TI.

What is the rationale for the use of 1kb windows for the calculation of the TI index?

1kb was used as the bin size due to read depth, particularly when looking at regions with lower amounts of H3.3 and when examining H3.1 turnover. Enhancers can be done in smaller windows. We have added Figure 3—figure supplement 1 which shows turnover at enhancers and super-enhancers when TI is calculated using a 300bp bin size (subsection “Time-ChIP identifies regulatory regions in ESCs”, third paragraph).

The use of linear regression slopes as the TI index raises concerns that the authors are assuming a linear relationship in histone turnover. What about regions that experience little turnover for long periods followed by spurts of high turnover?

We examine large numbers of stably pluripotent or differentiated cultures, so do not expect complicated behavior in the regions examined. We agree that a non-linear analysis might well be needed if one is doing an acute response; we do not have data that could be used to test this supposition though.

*How is the MACC calculated? There is insufficient information for the nature of this calculation, which is concerning since an entire section of the paper is based on this calculation. A reference to a submitted paper that was not provided is insufficient to judge the nature of the analysis and findings.*

We apologize for this oversight, we had thought the paper would be published by the time of submission and should have made the information available. The paper is now published and is cited:

Mieczkowski et al. (2016). MNase titration reveals differences between nucleosome occupancy and chromatin accessibility. Nature communications 7, 11485.

[Editors’ note: the author responses to the re-review follow.]

The manuscript has been improved but there are some remaining issues that need to be addressed before acceptance, as outlined below:

[…]

1) While the Discussion does in fact mention that Zhao came to the same conclusion on enhancers and cites the right paper, it comes too late in the manuscript. Hence, the Abstract makes an inappropriate novelty claim regarding high H3.3 turnover at regions known to be involved in gene regulation. Since Zhao has already shown this then they should acknowledge it in the Abstract and Introduction, i.e., "As previously shown, we found high H3.3 turnover at active regulatory regions including enhancers. However, we also find that even higher turnover occurs at super-enhancers […]".

We have edited the Abstract (“High turnover was seen at enhancers, as observed previously”) and the Introduction (“In ESCs, consistent with previous work, we found high H3.3 turnover at active enhancers (Ha et al., 2014; Kraushaar et al., 2013)[…]”) to refer to both the Kraushaar and Ha papers from Zhao’s group.